# Preparation of Humic Acid from Weathered Coal by Mechanical Energy Activation and Its Properties

**Xiujuan Feng [1,2,3,*], Rilong Xiao [1,2,3], Sékou Mohamed Condé [1,2,3], Chengliang Dong [1,2,3,*], Yanping Xun [4], Dalong Guo [4], Hui Liu [4], Kunpeng Liu [5] and Mingzhi Liang [5]**

[1]  School of Mines, China University of Mining and Technology, Xuzhou 221116, China; ts22020199p21@cumt.edu.cn (R.X.)
[2]  Industrial Technology Innovation Center for Ecological Restoration of Industrial and Mining Sites in the Petroleum and Chemical Industry, Xuzhou 221116, China
[3]  Mechano Chemistry Research Institute, China University of Mining and Technology, Xuzhou 221116, China
[4]  Inner Mongolia Environmental Governance Engineering Co., No. 7-1, Hailaer East Street, Xincheng District, Hohhot 010000, China; xunyanping@crcept.com (Y.X.); dalong_ustb@126.com (D.G.)
[5]  Xinjiang Jiangna Mining Industry Co., No. 1-105, Xingsheng Open Pit Coal Mine Office Building, Numaohu Mining District, Yiyu County, Hami 839304, China
*  Correspondence: xjfeng@cumt.edu.cn (X.F.); memail3000@126.com (C.D.)

**Abstract:** Humic acid (HA) is rich in functional groups with high activity, which can effectively improve the soil environment. The large reserves of weathered coal in China provide sufficient raw material guarantee for HA extraction and utilization. At present, the activation side of weathered coal is still the main technical difficulty that restricts HA extraction. In this study, the weathered coal from Inner Mongolia was used as the raw material, and the mechanical energy was used to activate the weathered coal through a planetary ball mill, which improved the extraction rate of HA and optimized the molecular structure and composition of HA. The effects of four parameters, namely, ball material ratio, ball milling time, ball milling speed, and ball size, on the free HA content of weathered coal were investigated, the HA was extracted by alkaline extraction method, and the activated weathered coal and the extracted HA were characterized. The results showed that a ball material ratio of 9:1, a ball milling speed of 200 r/min, a ball milling time of 200 min, a milling ball size of $\Phi5:\Phi10:\Phi15 = 48:42:45$ and $56:42:37$ are the optimal parameters for the mechanical energy activation, and the HA extraction rate of activated weathered coal under these conditions reached 82.3%, which was 15% higher than that of the unactivated one. Moreover, the aroma of the ball-milled weathered coal increased, the content of oxygenated functional groups increased, and the molecular weight and aroma of HA increased. This provides scientific theoretical guidance for the preparation of HA with high aromaticity and large molecular weight from weathered coal.

**Keywords:** high-energy ball milling; mechanical energy; weathered coal; activation; alkali dissolution; acid precipitation method; humic acid

## 1. Introduction

Humic acid (HA) is an organic substance rich in functional groups, with a complex structure and variable molecular weight [1], which can be used in a variety of fields, such as forestry, agriculture, water treatment, medical care, etc. [2]. The structural properties of HA determine its applications, among which the molecular weight [3] and aromaticity [4] of HA are two important characteristics. Small molecular weight HA has better water solubility and stronger complexation with metals and thus can be used in wastewater treatment [5], while large molecular weight HA is more suitable for soil improvement due to its poor water solubility and inadvisable loss [6]. Chen [7] found that black soil HA is

suitable for soil remediation because of its high aromaticity and flocculation, which is favorable for the accumulation of organic matter and easier to react with metal cations.

It is well known that weathered coal has low calorific value and is not easy to burn but is rich in HA, so the extraction of HA from weathered coal as a raw material is one of the most efficient ways to utilize it as a non-fuel product. Weathered coal, as a kind of coal companion, has huge reserves. China's weathered coal reserves of about 10 billion tons, are distributed in Xinjiang, Heilongjiang, Shaanxi, Inner Mongolia, Yunnan, and other places [8]. Moreover, China has a large number of coal mines, so the amount of weathered coal produced each year is extremely large, and a large amount of weathered coal is in urgent need of development and utilization. Typically, the total HA content of weathered coal ranges from 30 to 70%, with higher levels reaching more than 70% [9]. Weathered coal with high total HA content can be utilized directly, e.g., for soil improvement, wastewater treatment, radioactive waste disposal [10], etc. However, for most of the weathered coal, due to its low total HA content leading to the direct use of the effect is not ideal, and it may also contain heavy metals and other hazardous substances, the direct use of which may cause secondary pollution, and therefore it is necessary to extract the HA in the weathered coal through certain technical means and then apply it. For weathered coal, the level of its internal total HA and free HA content is the key to its utilization. Total HA includes free HA, and it is generally unlikely to increase the total HA content, but it is feasible to increase the free HA content. At present, increasing the free HA content in coal is called coal activation, and the activation technology includes ultrasonic activation [11], acid activation [12], alkali activation [13], and so on. However, the disadvantages of these techniques are obvious, such as poor ultrasonic activation, acid activation, and alkaline activation, which require a large amount of chemicals and serious environmental pollution. Therefore, an activation process with good treatment effect, a green environment effect, and fast speed is necessary.

Mechanical force chemistry is defined as the description of chemical and physicochemical transformations of substances during aggregation induced by mechanical energy, and it has multiple advantages, such as process simplicity, ecological safety (no solvents, melting operations, etc.), and the possibility of making a product in the metastable state, which is difficult (or impossible) to obtain using other conventional methods [14]. The use of mechanical forces can be traced back to the use of friction to create fire, and since then it has been used in extractive metallurgy, crystal engineering, materials engineering, the coal industry, the construction industry, agriculture, pharmaceuticals, and waste treatment. The technology covers many important reactions such as faster decomposition and synthesis [15], grafting [16], and polycrystalline transformations [17]. Mechanical forces have potential applications in pollution remediation and waste management for fly ash, coal gangue, and weathered coal, etc. [18]. The mechanochemical reactions involved include redox [19], polymerization [20] and polymer rearrangement [21], recrystallization [22], dehydration [17], etc. Heavy metals and other toxic and hazardous substances may be present in weathered coal [23,24], so the use of mechanical force activation of weathered coal can reduce the risk of its harmful effects on environmental pollution while realizing the improvement of the HA extraction rate and optimization of performance. Several studies have shown that activation of weathered coal using mechanical energy increases the yield of HA and the number of reactive functional groups [25] and that the aromatic carbon weight yield of C-O in HA obtained by extraction of mechanically activated weathered coal increases [26]. In addition, the existing weathered coal activation techniques tend to obtain small molecular weight HA; however, it has been shown that mechanical force can initiate the polymerization reaction [27], so the use of mechanical force activation of weathered coal can also obtain larger molecular weight HA, which at the same time provides a new way of thinking for the preparation of high molecular weight HA.

The purpose of this study is to improve the properties of weathered coal by using mechanical energy activation to increase its free HA content and then to increase the HA

yield. HA of larger molecular weight and higher aromaticity is obtained from weathered coal by using the action of mechanical energy.

## 2. Materials and Methods

### 2.1. Materials

Materials included potassium hydroxide (KOH, AR), deionized water, hydrochloric acid (HCl, 20%), sodium hydroxide (NaOH, AR), and sodium pyrophosphate ($Na_4P_2O_7$, AR). The above reagents were purchased from Shanghai Titan Technology Co., Ltd., Shanghai, China. Planetary Vertical Ball Mill, model QXQM-2 (Changsha, China) is made of stainless steel, and the grinding balls are steel balls.

The raw materials of weathered coal for the test were from Inner Mongolia, its composition is shown in Table 1, where the total HA content in weathered coal was determined by volumetric method with $Na_4P_2O_7$ ($Na_4P_2O_7$ can cause the insoluble HA fixed by Ca, Mg, etc. to be dissolved in the form of sodium HA) and NaOH, and the free HA content was determined by NaOH.The elemental analysis result shown in Table 2, the H/C, O/C, N/C, and S/C represent the respective atomic ratio. In order to ensure the accuracy of the test results, a total of three tests were taken to determine the average value.

Based on the results of industrial and elemental analysis tests of weathered coal, the structural parameters of weathered coal can be further analyzed and calculated. This method assumes that the C element includes only aliphatic carbon and aromatic carbon and can be used for analytical comparisons between weathered coals. The aromaticity is $f_a = (100 - V_{daf}) \times 0.9677/C_{daf}$; the larger the value represents the larger the aromaticity. The number of rings of individual C atoms is $(R/C)_u = 1 - 0.5f_a - 0.5(H/C)$; the larger the value, the larger the degree of aromatization of the coal when substituting the relevant data $f_a = 0.89$ and $(R/C)_u = 0.174$, respectively.

**Table 1.** Analysis of weathering coal components.

| Component | Moisture ($M_{ad}$) | Ash Content ($A_d$) | Volatile Matter ($V_{daf}$) | Fixed Carbon ($FC_{ad}$) | Total HA | Free HA |
|---|---|---|---|---|---|---|
| Content (%) | 6.10 | 43.56 | 45.51 | 32.36 | 31.9 | 25.8 |

**Table 2.** Elemental analysis of weathered coal.

| Element | N | C | H | S(%) | O | H/C | O/C | N/C | S/C |
|---|---|---|---|---|---|---|---|---|---|
| Content (%) | 0.88 | 58.30 | 3.067 | 0.209 | 37.544 | 0.631 | 0.483 | 0.129 | 0.001 |

### 2.2. Mechanical-Energy-Activated Weathered Coal

The weathered coal after mechanical energy activation was prepared by the following methods and steps. Take the appropriate amount of weathered coal before grinding and crushing in the mortar. After the 80-mesh sieve, dry in the oven for 24 h. The dried weathered coal and the appropriate amount of steel balls are added to the ball mill tank. The mass ratio of weathered coal and steel balls is 1:3~1:18, and the steel ball specifications for the Φ5:Φ10:Φ15 = 24:42:69~72:42:21 (Φ5 means that the diameter of the ball for the 5 mm, 24:42:69) (Φ5 means the diameter of steel ball is 5 mm; 24:42:69 means the mass ratio). Start the ball mill, set the ball mill speed to 50~300 rpm, and set the ball mill time to 50~300 min. After the ball mill is finished, the weathered coal is obtained after mechanical energy activation with different parameters.

### 2.3. HA Extraction

In order to fully extract HA, the following methods and steps which were used for extraction from weathered coal were shown in Figure 1. Accurately weigh 1 g of weathered coal in a 200 mL conical flask, add the concentration of 2%, the volume of 13 mL of potassium hydroxide solution, and then put the conical flask into the rotational

speed of 80 rpm constant-temperature water bath shaker reaction. Select a temperature of 55 °C to maintain the reaction of the weathered coal and potassium hydroxide solution for 5 h. The end of the reaction, namely the black-brown solution containing HA, was poured out of centrifugal tubes and sealed, with a centrifuge set at 4000 rpm and centrifugal time of 30 min to purify the HA solution containing impurities by centrifugation. After the reaction, the black-brown solution containing HA was poured into a centrifuge tube and sealed well, and the HA solution containing impurities was purified by centrifugation with a centrifuge set at 4000 rpm and a centrifugation time of 30 min. After centrifugation, the solid–liquid separation was carried out by taking the upper layer of the clear solution in a 50 mL beaker, adjusting the pH of the HA solution to 2, with dilute hydrochloric acid at a concentration of 2% and a pH meter, and separating it with a sand-core filter after the HA was fully precipitated. After filtration, the HA solids together with the filter membrane were dried in an oven at 50 °C for 12 h. The total mass of the dried HA and the filter membrane was recorded and then the HA solids on the filter membrane were separated and weighed to record the mass of the filter membrane, and, finally, the HA solids obtained were sealed in a sample bag and placed in a dry and cool place for storage.

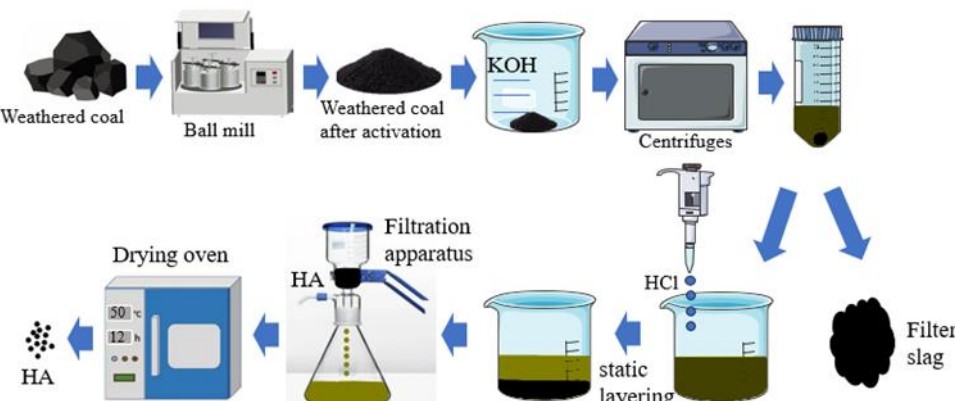

**Figure 1.** Flow chart of humic acid preparation from weathering coal activated by mechanical force.

### 2.4. Characterization of the Properties of Weathered Coal and HA

(1) Scanning electron microscope (SEM) was used to characterize the weathered coal before and after mechanical energy activation. In this paper, a Quatto S field emission scanning electron microscope-energy spectrometer manufactured by FEI Company, Czech Republic (Karlsruhe, Germany), was selected to observe the microscopic morphology of the materials. For sample preparation, the samples were placed in anhydrous ethanol, ultrasonicated, and then uniformly coated on conductive adhesive tape, then sprayed with gold and tested under 20 kV accelerating voltage.

(2) X-ray diffraction (XRD) was used to characterize weathered coal before and after mechanical energy activation. In this paper, the D8 Advance X-ray diffractometer produced by Bruker AXS Co., Ltd. (Karlsruhe, Germany) was used to test the materials before and after the mechanical force chemical action and process the test data through MDI jade6 software which is used for material analysis and crystallography research to analyze the composition of raw materials and the changes in the crystal structure of the materials before and after the mechanical force chemical action. In terms of sample preparation, the material was ground and dried through a 200-mesh sieve, and 0.5 g of the sample was placed in the X-ray diffraction sample tank, compacted and smoothed, and then analyzed on the machine under the test conditions of a Cu target (K$\alpha$ radiation), an operating voltage of 40 KV, a current of 100 mA, a scanning range of 10–80°, and a scanning speed of 2°/min.

(3) Fourier infrared spectroscopy (FT-IR) was used to characterize the weathered coal before and after mechanical energy activation. In this paper, the TENSORII Fourier infrared spectrometer from Bruker Spectroscopy Instruments (Karlsruhe, Germany) was used to test the materials and process the test data through OMNIC software version 9.2 which is used primarily for spectral data analysis in various fields such as chemistry, materials science, and pharmaceuticals to analyze the changes of chemical bonds and functional groups of the materials before and after activation. For sample preparation and testing, the material to be tested was ground and dried through 200 sieves, and 30 mg of the sample was mixed with 4.5 g of dried potassium bromide powder and pressed, and the samples to be tested were placed in the sample stage and then tested and analyzed by the machine, with the range of the wave number of the test being selected as 500–4000 cm$^{-1}$, the resolution of the wave number of the test being 1.0 cm$^{-1}$, and the number of the scanning times being 33 times.

(4) The content of oxygen-containing functional groups in weathered coal was determined according to the soil organic matter research method [28]. The total acidic group was determined by the barium ion exchange method, the carboxyl group was determined by the calcium acetate solution method, and the phenolic hydroxyl content was calculated by the difference subtraction method, i.e., phenolic hydroxyl content = total acidic group content − carboxyl group content.

(5) The $E_4/E_6$ values of HA were determined by ultraviolet spectrophotometer (UV-VIS) (Shanghai, China). In total, 0.02 g of HA was accurately weighed and dissolved in 80 mL of 0.05 mol/L NaHCO$_3$ solution, and the pH of the solution was adjusted to 8.0 with NaOH or HCl, and then 0.05 mol/L NaHCO$_3$ solution and 100 mL volumetric flasks were used to formulate a solution with a concentration of HA of 200 mg/L. The 0.05 mol/L NaHCO$_3$ solution was taken as the reference solution. The NaHCO$_3$ solution of 0.05 mol/L was used as the reference solution, and the average value was calculated by taking data three times during the test; the calculation method is as follows:

$$E_4/E_6 = E_{465}/E_{665}$$

$E_{465}$ is the absorbance of the solution at a wavelength of 465 nm;
$E_{665}$ is the absorbance of the solution at a wavelength of 665 nm.

(6) ICP-OES was used to test the activated weathered coal HA for possible metal elements. ICP-OES (Connecticut, USA) of the PerkinElmer 8300 model was used for the testing of HA. A suitable amount of HA was dissolved in 100 mL of 0.05 mol/L NaHCO$_3$ solution before the test, and then the test was performed.

## 3. Results and Discussion

### 3.1. Effect of Mechanical Energy on Free HA Content in Weathering

The effect of mechanical energy on the free HA content in weathered coal was explored by a one-way test. The effects of different ball material ratios, ball milling time, ball milling speed, and milling ball size were investigated separately. The free HA content in weathered coal after activation of each ball milling parameter is shown in Figure 2, and the following results were obtained. When the ball material ratio is 9:1, the ball milling speed is 200 rpm, the ball milling time is 200 min, and the grinding ball size is Φ5:Φ10:Φ15 = 48:42:45 and 56:42:37 (the mass ratio of the three sizes of steel balls). The free HA content of weathered coal after activation reaches a maximum of 28.9%, which is 3.1% higher than the original free HA content.

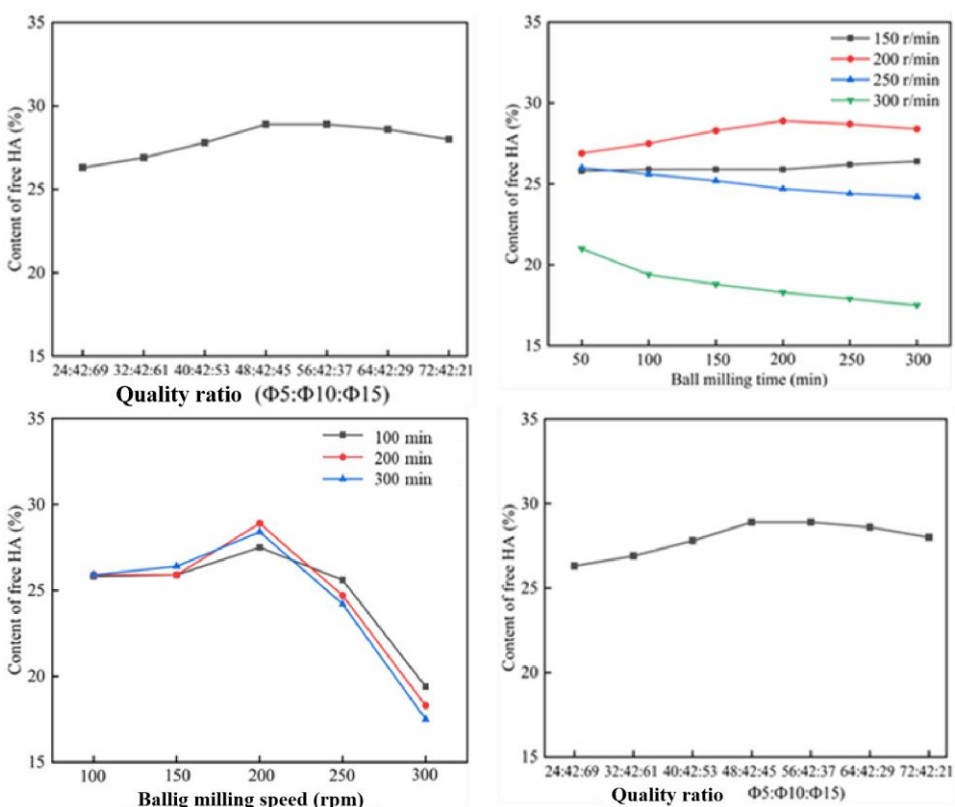

**Figure 2.** Effect of different ball milling parameters on the content of free HA.

### 3.2. Extraction of HA

Humic acid extraction is shown in Table 3. The HA extraction of activated weathered coal was 67.3%, and the HA extraction of weathered coal activated by mechanical energy was 82.3%. In order to highlight the positive effect on HA extraction after mechanical energy activation of weathered coal, the results of other HA extraction studies are compared here.

**Table 3.** Comparison of extraction of HA from different raw materials with different extraction methods.

| Raw Materials | Extraction Methods | Extraction of HA | Reference |
|---|---|---|---|
| Lignite | Nitric acid oxidation | 57.8% | [29] |
| Lignite | Penicillium ortum MJ51 | 63.9% | [30] |
| Lignite | Hydrothermal treatment | 68% | [12] |
| Weathered coal | $Fe_3O_4/LaNiO_3$ catalyst treatment | 48.5 | [31] |
| Peat | Ultrasound treatment | 60% | [32] |
| Weathered coal | Non-activated extraction | 67.3% | This study |
| Weathered coal | Activated by mechanical energy | 82.3% | This study |

Oxidative treatment is a more applied coal activation method, and the effect of HA extraction by oxidative method has a high enhancement; for example, the HA extraction reached 57.8% after activation by nitric acid, the HA extraction reached 48.5% by oxidative extraction using catalyst ($Fe_3O_4/LaNiO_3$), and the HA extraction reached 68% by hydrothermal treatment. However, the shortcomings of the above oxidation methods are too obvious: one is the high consumption of chemicals and low extraction of the nitric acid activation method; the second is the catalyst oxidation method with high requirements on the reaction environment; and the third is the high energy consumption of hydrothermal treatment, as the extraction is still not high, and it cannot be realized for large-volume

preparation and application. In addition, in order to further improve the extraction of HA from coal, microbial extraction methods have been developed, such as Penicillium ortum MJ51, which has achieved a 63.9% extraction of HA, but the effect is still not obvious. Upon comparison with others' studies, the mechanical force activation treatment of weathered coal and the alkaline dissolution and acid precipitation extraction method of HA adopted in this study have significant advantages of short process speed, simplicity, high efficiency, low environmental pollution, low energy consumption, and high HA extraction.

### 3.3. SEM, XRD, FT-IR Characterization and Other Test Results of Weathered Coal

(1) The SEM characterization results are shown in Figure 3. Compared with the pre-activation ones, the particle size of weathered coal particles after mechanical activation was obviously reduced; the size specification of each particle size was more uniform and homogeneous; and the isometric aspect of the particles reduced, presenting laminated and flaky structures; the roughness of the particle surface increased; the specific surface area increased; the surface of the particles had a lot of stacked flakes; and more crevices existed.

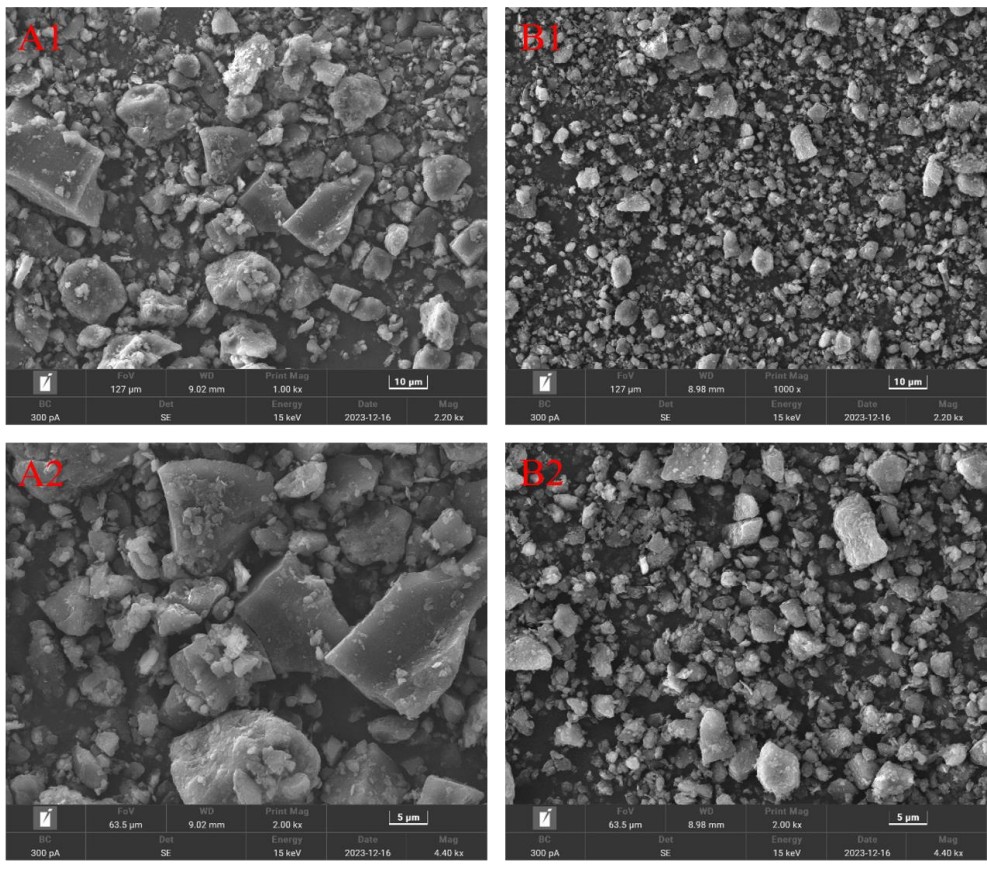

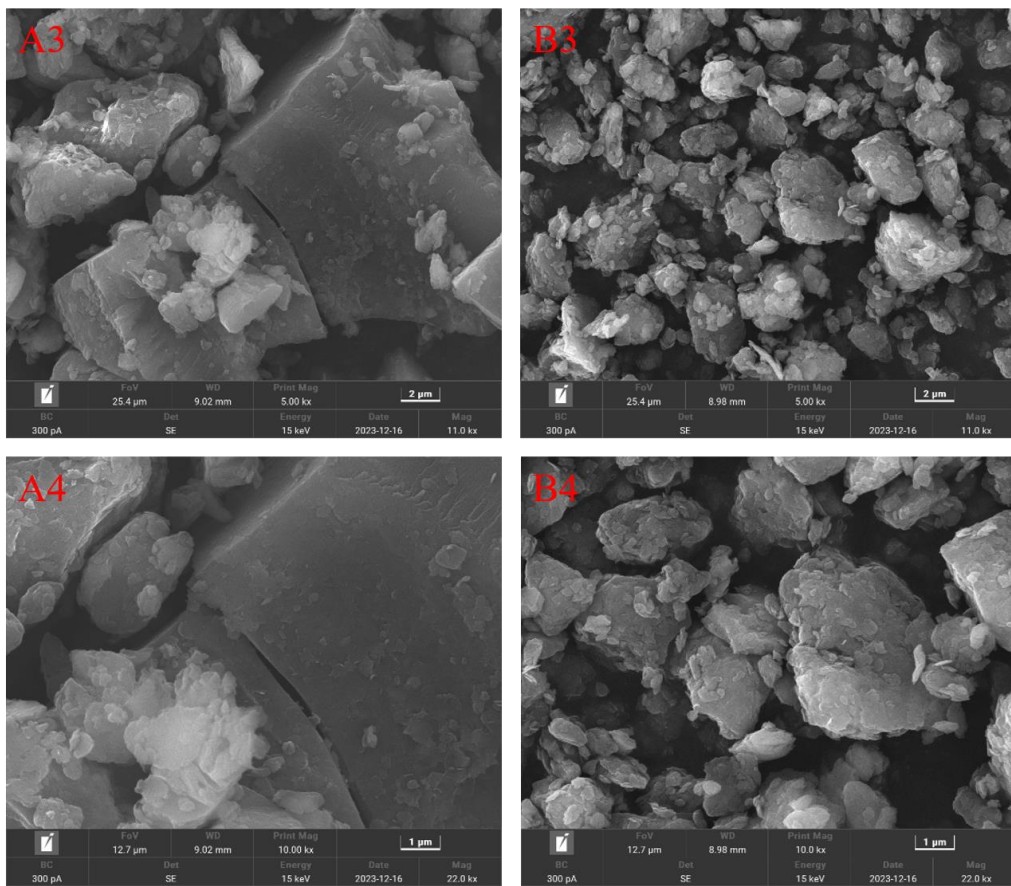

**Figure 3.** SEM characterization results of weathered coal ((**A**) represents weathered coal before activation, (**B**) represents weathered coal after activation. 1, 2, 3, and 4 represent magnification of 1000 times, 2000 times, 5000 times, and 10,000 times, respectively. For example, A1 represents the microscopic images of weathered coal before activation, and so on for the rest).

(2) The correlation indexes of the aromaticity of weathered coal before and after activation are shown in Table 4. It was found that the H/C value of the weathered coal after activation was reduced, and the aromaticity $f_a$ and the ring number of individual C atoms $(R/C)_u$ were increased, which indicated that the aromaticity increased after mechanical activation. $(R/C)_u$ increased, which indicated an increase in aromatic groups and aromaticity of weathered coal after mechanical activation. With the ball milling, the temperature is increasing, according to "Hilt's law"; the increase in coal rank with depth is due to the increase in temperature with depth, that is to say, it can be considered that the coal rank increases with the increase of temperature, and the higher the rank is, the higher the aromaticity of the coal is [33], so it can be considered that the change in the weathered coal after the ball milling is similar to the change in the rank to a certain extent, which can be regarded as the change in the coal rank. Therefore, it can be assumed that the change in weathered coal after ball milling is similar to the change in coal rank to a certain extent and degree, which can explain why there is an increase in the aromaticity of coal after ball milling.

**Table 4.** Related indexes of aromatic degree of weathered coal before and after activation.

| Weathered Coal | H/C | $f_a$ | $(R/C)_u$ |
|---|---|---|---|
| Pre-activation | 0.631 | 0.89 | 0.174 |
| Activated | 0.613 | 0.93 | 0.220 |

(3) The XRD characterization results are shown in Figure 4. The main minerals present in the weathered coal are silica and kaolinite, and both the weathered coal before and after activation show high background intensities, suggesting that both contain a certain proportion of highly disordered material in the form of amorphous carbon [34]. The peak with a diffraction angle of 26° corresponds to microcrystals in polycondensed aromatic rings, which are related to the stacking between aromatic rings, and the intensity of the peak at this location of the weathered coal increases after mechanical force, indicating an increase in aromaticity; except for that, mechanical force did not lead to any significant changes in the XRD patterns of the weathered coal.

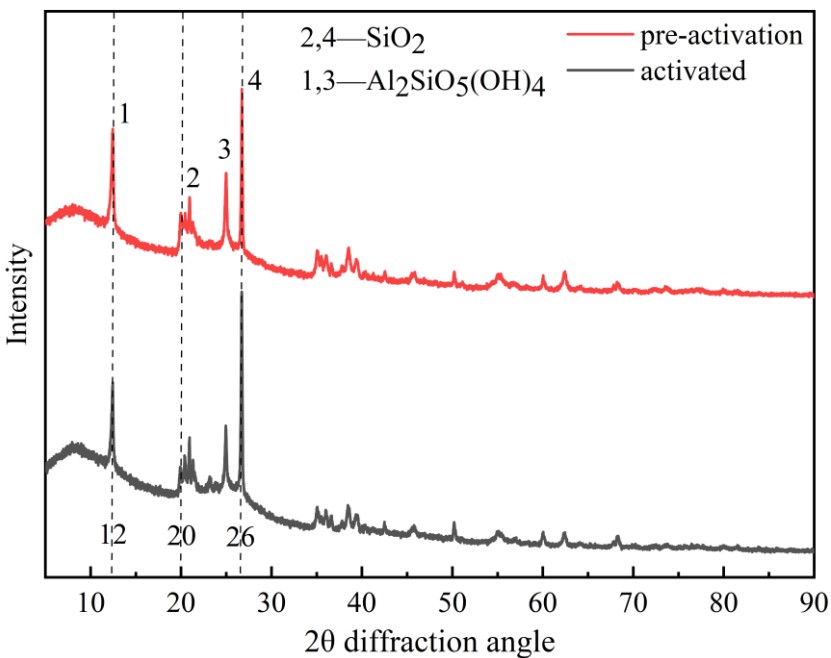

**Figure 4.** XRD spectra of weathered coal.

(4) The results of FT-IR characterization are shown in Figure 5. Each homing region of the infrared characteristic peaks of weathered coal is as follows: kaolinite (-Si-O-H-) vibrations at 3730–3610 $cm^{-1}$; hydrogen bonding -OH, -NH, -NH$_2$ telescoping vibrations at 3600–3100 $cm^{-1}$; aromatic CH telescoping vibrations at 3100–3000 $cm^{-1}$; hypromellitic CH$_2$ symmetric telescoping vibrations (with a small amount of methyl-CH$_3$) at 2970–2900 $cm^{-1}$ CH$_2$ asymmetric stretching vibration at 2880–2800 $cm^{-1}$ (with a small amount of methyl-CH$_3$); -COOH, -CO vibration at 1790–1680 $cm^{-1}$; carboxylate, aromatic C=C double bond, hydrogen-bonded carbonyl-C=O- at 1670–1530 $cm^{-1}$; aromatic nucleus, aromatic C=C double bond vibrations; methyl-CH$_3$ in-plane bending vibrations at 1580–1400 $cm^{-1}$; alkyl, methyl -CH$_3$ in-plane bending vibrations at 1390–1350 $cm^{-1}$; lipids at 1330–1100 $cm^{-1}$; hydroxyls, ketones, acetals at 1220–1070 $cm^{-1}$; clay minerals, such as kaolinite, at 1050–970 $cm^{-1}$; and Si-O- vibrations at 1330–1480 $cm^{-1}$; Si-O- vibration at 1050–970 $cm^{-1}$; quartz mineral (-Si-O-) vibration at 920–900 $cm^{-1}$; one hydrogen atom out-of-plane deformation vibration on the aromatic nucleus at 890–820 $cm^{-1}$ (class I hydrogen atom); two neighboring hydrogen atoms out-of-plane deformation vibration on the aromatic nucleus at 810–790 $cm^{-1}$ (class II hydrogen atom); four neighboring hydrogen atoms out-of-plane deformation vibration on the aromatic nucleus at 755–745 $cm^{-1}$ (class II hydrogen atom); four neighboring hydrogen atoms out-of-plane deformation vibration on the aromatic nucleus at 1220–1070 $cm^{-1}$ (class II hydrogen atom); (class IV hydrogen atom); 710–695 $cm^{-1}$ is a benzene ring folding vibration; and 595–420 $cm^{-1}$ is an inorganic mineral.

After ball milling, the peak intensities at 1600 $cm^{-1}$, 797 $cm^{-1}$, 754 $cm^{-1}$, and 694 $cm^{-1}$ were increased at four places, and the combination of the test results of $f_a$ and $(R/C)_u$ values of the weathered coal and the XRD test results showed that the increase in the aromaticity

of the weathered coal after activation by the optimal mechanical force activation process was due to the stacking and condensation of the various aromatic carbons in the weathered coal under the action of the compressive stress to form a new and larger aromatic layer, which was formed by the stacking and condensation of the aromatic carbons in the weathered coal under compressive stress [33].

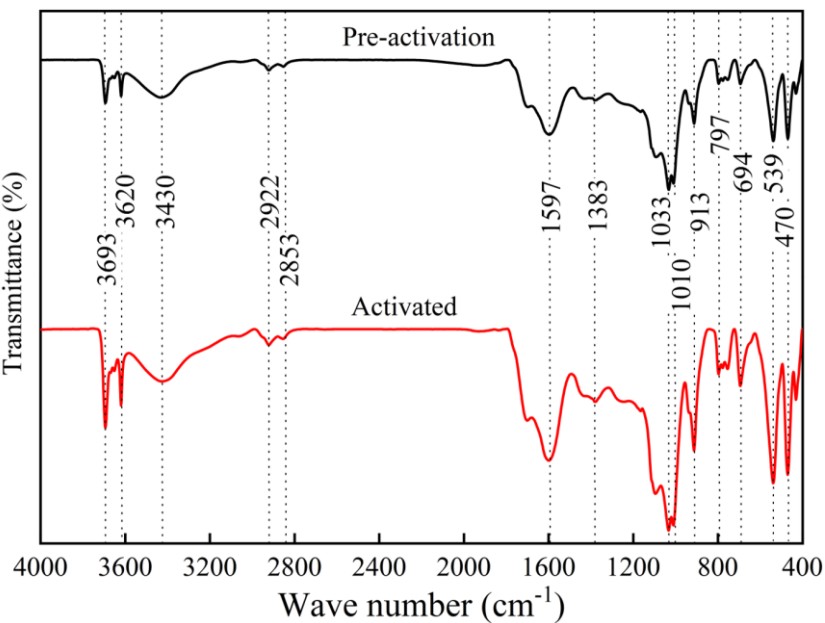

**Figure 5.** FT−IR spectra of weathered coal.

(5) The content of oxygen-containing functional groups of weathered coal is shown in Table 5. In addition, as shown in Figure 4, the intensity of the absorption peaks of each oxygen-containing functional group of the activated weathered coal increased, which, combined with the measurement results, indicates that the content of the total acidic, carboxyl, and phenolic hydroxyl groups of the activated weathered coal increased. This is because, due to the oxidation reaction accompanying the process of ball milling, the content of phenolic hydroxyl groups increased due to the fracture of some of the aryl ether bonds contained in the weathered coal, and at the same time some hydroxyl groups were oxidized to form new carboxyl groups, leading to an increase in carboxyl groups [35]. Some of the hydroxyl groups were oxidized to form new carboxyl groups, resulting in an increase in carboxyl group content [36].

**Table 5.** Oxygenated functional group content of weathered coal.

| Weathered Coal | Total Acidic Group Content (mmol/g) | Carboxylic Acid Content (μmol/g) | Phenol Hydroxyl Content (mmol/g) |
|---|---|---|---|
| Pre-activation | 2.89 | 282 | 2.61 |
| Activated | 3.23 | 454 | 2.77 |

(6) Summary: Effects of Mechanical Forces on Weathered Coal

First of all, it is physical action. In the ball milling process, under the joint action of compressive stress and shear stress, the weathered coal deformation occurs first. In this stage, the surface of the weathered coal is constantly being impacted by the steel ball, the other weathered coal, and the ball milling tank wall, resulting in the edges of the weathered coal are smoothed, appearing obtuse, which has the greatest impact of the steel ball, which is due to the highest density, the hardest, and the largest contact area with the weathered coal. When the weathered coal deformation to a certain extent and then the rupture occur, the internal material exposure, originally fixed by the minerals of the HA,

can be exposed and the mechanical force so that the link between the two is broken, resulting in the release of HA and improving the free HA content. With the large particles of weathered coal broken into small particles of weathered coal, the overall particle size decreases. At this time, due to the reduction in particle size, the particles move in the gap between the steel ball, the tank wall, and the particles of the three, and the coal weathered by the steel ball impact frequency decreases, so the increase in the number of small balls is conducive to improving the efficiency of grinding. Due to the complex composition of weathered coal, the density of each region is inconsistent; in the weathered coal rupture, its particles are instantly cracked into many small particles of different sizes, and the interface formed on the rupture side will have many cracks and grooves, resulting in a surface that is relatively rough. At the same time, many smaller particles, due to the constant extrusion and collision, led to the process of movement by the particles of the interface of the roughness of the capture, and the extrusion makes the two by the combination of the two particles much closer, so it is observed that there are other small particles attached to the surface of the weathered coal particles after ball milling.

The second is chemical action. Under the action of strong mechanical force, the crystalline water inside the weathered coal is precipitated and transformed into adsorbed water, which exacerbates the agglomeration between the particles. The grain size of weathered coal decreases rapidly due to impact, extrusion, and friction by steel balls, etc. At the same time, a large number of crystal defects and lattice deformations also form, and more and more substances are transformed from crystalline to highly disordered and amorphous forms. Mechanical force breaks the organic macromolecules in weathered coal and increases the free hydroxyl groups. The mechanical force process is accompanied by an oxidation reaction; part of the benzene ring is oxidized to open the ring to produce fatty acids, and at the same time, under the action of compressive stress, the aromatic carbon will also be stacked and condensed to form a new, larger aromatic layer.

### 3.4. Performance Analysis of HA

(1) The results of UV-VIS characterization of HA are shown in Figure 6. The E4/E6 value of HA is commonly used in the chemical industry to characterize the aromaticity and molecular weight of HA because of its complex structure, and the smaller the value is, the lower the aromaticity and molecular weight are. The E4/E6 value of the weathered coal HA extract before mechanical activation was 4.125, and it can be seen that the lowest point of E4/E6 in the graph was 4.004, which corresponded to a ball milling speed of 200 rpm and a ball milling time of 200 min. It is worth noting that the aroma and molecular weight of HA decreased faster with the increase in time at the speed of 300 rpm, which is presumed to be due to the high mechanical energy crushing the polycondensation of HA. The mechanical energy crushed the condensed HA, the macromolecules were broken up and decomposed, and the originally stacked aromatic layer was destroyed by the tangential force [37].

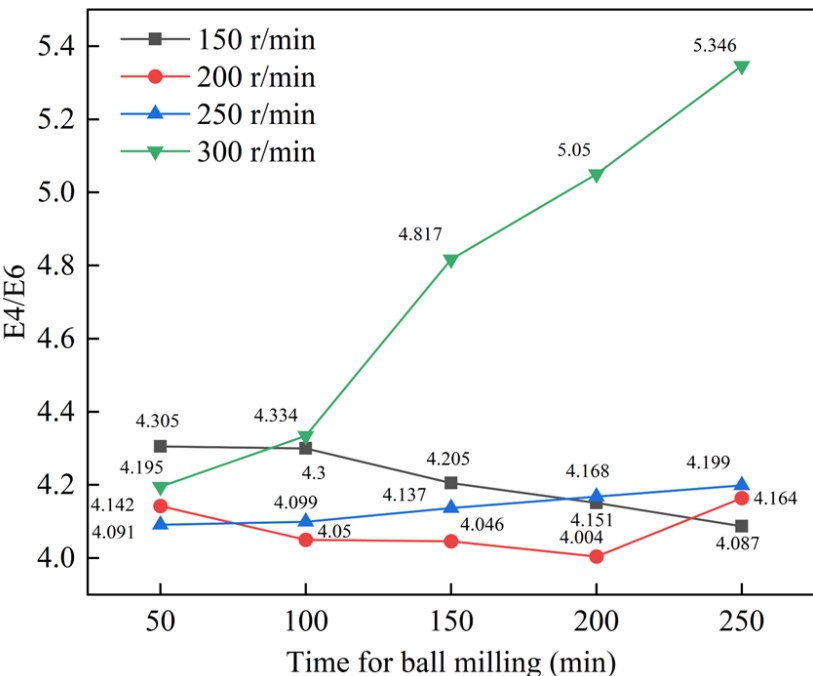

**Figure 6.** E4/E6 of humic acid of weathering coal under different activation parameters.

(2) The infrared spectral results of HA [38,39] are shown in Figure 7. The vibration peak at 3434 cm⁻¹ is a large number of free and hydrogen bonded OH groups; the peak intensity is high, indicating that the HA is rich in hydroxyl groups; and the intensity of the peaks of the two HAs is basically the same: 3000–2700 cm⁻¹ belongs to the fat region, the peaks at 2922 cm⁻¹ and 2850 cm⁻¹ are CH₂ asymmetric and symmetric telescopic vibration, respectively, and the peaks of the two HAs are not shifted and the intensity is basically the same; 1950–1800 cm⁻¹ is the carbonyl C=O vibration, presumably a carboxylic acid or ketone, and the intensity of the peaks of the activated HA is stronger than that of the activated one. The peaks at 2922 cm⁻¹ and 2850 cm⁻¹ were CH₂ asymmetric and symmetric telescopic vibration, respectively, and the two HAs did not have a large degree of peak shift, and the intensity was basically the same; the C=O vibration of the carbonyl group at 1950–1800 cm⁻¹ was presumed to be a carboxylic acid or ketone, and the intensity of the peaks in this area of the HA was stronger than that in the one before activation. The C=O vibration of the carboxylic acid group at 1708 cm⁻¹ was a stronger intensity of absorption in this area of the HA than that in the one before activation, which indicated that the content of the carboxylic acid functional groups of the HA was greater in the one extracted from the one after activation than that in the one before activation. The C=C vibration of aromatic was at 1602 cm⁻¹, with higher intensity after activation, which is consistent with the results of UV-VIS spectral analysis of HA in 4.5.2; the methylene deformation vibration and methyl symmetry deformation vibration was at 1422 cm⁻¹ and 1377 cm⁻¹, respectively, which are not obvious in terms of the peak intensity and the changes before and after activation. The C=O vibration was at 1231 cm⁻¹, which is presumed to be an Aromatic ether; 757 cm⁻¹ for the four adjacent hydrogen atoms on the aromatic nucleus out-of-plane deformation vibration, both of the peak intensities there were basically the same, namely 907 cm⁻¹ and 476 cm⁻¹ peaks for the minerals, both of which are very weak in these two peaks and much lower than the intensity of the peaks of the weathered coal in these two places. It can be seen that the extracted HA is very low in minerals, and the purity of the extracted HA is relatively high.

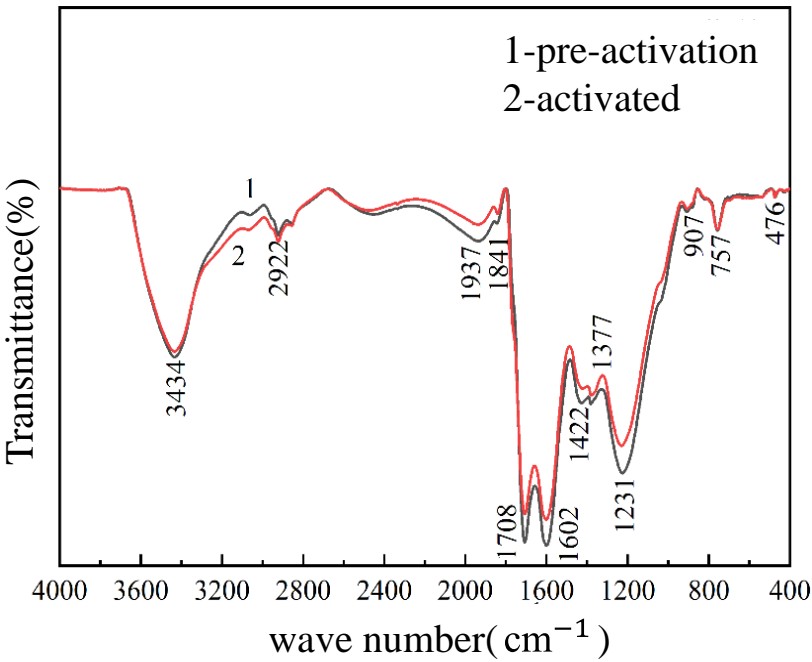

**Figure 7.** Infrared spectrum of humic acid of weathering coal before and after activation.

(3) The results of the ICP-OES analysis of HA are shown in Table 6. From the analytical results, it can be seen that the content of each metal element in HA obtained from weathered coal extraction before and after activation is very low; for example, the content of the Cd element is zero, which indicates that HA extracted from this study is of high purity.

**Table 6.** Content of some metallic elements in humic acid extracted from weathering coal before (percentages are relative deviation).

| Element | Content (μg/mL) (before Activation) | Content (μg/mL) (after Activation) |
| --- | --- | --- |
| Mn | 0.0397 (2.10%) | 0.0367 (1.01%) |
| Cu | 0.0414 (0.13%) | 0.0398 (0.11%) |
| Cd | 0 (2.68%) | 0 (4.33%) |
| As | 0.0062 (3.54%) | 0.0064 (2.68%) |
| Mg | 0.0072 (1.75%) | 0.0036 (3.41%) |
| Ca | 0.0107 (0.81%) | 0.0088 (0.57%) |
| Cr | 0.0051 (2.41%) | 0.0062 (3.31%) |
| Al | 0.0074 (0.98%) | 0.0079 (2.64%) |
| Fe | 0.0186 (2.06%) | 0.0121 (3.11%) |
| Pb | 0.0069 (0.75%) | 0.0045 (1.03%) |
| Zn | 0.0088 (2.43%) | 0.0084 (1.36%) |

## 4. Conclusions

(1) Weathered coal activated by mechanical energy has reduced particle size, increased specific surface area, and increased aroma. The free HA content was 28.9%, which was 3.1% higher than that in the unactivated weathered coal. The optimal process is a ball material ratio of 9:1, a ball milling speed of 200 rpm, a ball milling time of 200 min, and a milling ball inch size of Φ5:Φ10:Φ15 = 48:42:45 and 56:42:37.

(2) Under the same extraction method, the HA extraction rate of weathered coal activated by mechanical energy is up to 82.3%, which is 15% higher than that of unactivated weathered coal. The aroma and molecular weight of HA are significantly increased, the content of functional groups is increased, the metal content is extremely low, and the performance is better.

**Author Contributions:** Conceptualization, X.F.; methodology, X.F. and R.X.; investigation, X.F., R.X., S.M.C., Y.X., D.G., H.L., K.L., M.L. and C.D.; validation, X.F., R.X. and C.D.; formal analysis, X.F., R.X., S.M.C. and C.D.; resources, X.F.; data curation, X.F. R.X., S.M.C. and C.D.; writing—original draft, X.F., R.X. and S.M.C.; writing—review and editing, X.F. and R.X.; visualization, R.X. and S.M.C.; supervision, X.F., Y.X., D.G., H.L., K.L., M.L. and C.D.; project administration, X.F. and C.D.; funding acquisition, X.F. All authors have read and agreed to the published version of the manuscript.

**Funding:** The work was funded by the Major Innovation Program of Shandong Province (2021CXGC011206); National Natural Science Foundation of China (Key Program, 52130402); the National Key R&D Program of China, grant number 2021YFC2902100.

**Data Availability Statement:** Data are contained within the article.

**Conflicts of Interest:** Mr. Yanping Xun, Mr. Dalong Guo and Dr. Hui Liu are employees of Inner Mongolia Environmental Governance Engineering Co. Mr. Kunpeng Liu and Mr. Mingzhi Liang are employees of Xinjiang Jiangna Mining Industry Co. There are no conflicts to declare.

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
