# Peer review of "Preparation of Humic Acid from Weathered Coal by Mechanical Energy Activation and Its Properties"

_minerals, doi:10.3390/min14070648_

Round 1

Reviewer 1 Report

Comments and Suggestions for Authors

The subject of the article is very interesting. I was investigated the preparation of humic acid from weathered coal applying mechanical energy by a milling process.

However,  the entire text  needs to be revised and written in a more formal way.

For instance, Table 1 is not associated with the text. The same for Figures 2, 3, 4 and 5. Section 2.2 and part of the sections 2.3 and 2.4 are written like a recipe. They need to be improved.

Two questions:

Why didn't the authors associate the production of humic acids with the work index?

What is the particle size distribution after each grinding process? Is it associated with the increase in humic acid?

The conclusion needs to be improved, confirming or not the hypothesis that gave rise to the work. 

Comments on the Quality of English Language

English is not my native language, but editing of english language is required.

Reviewer 2 Report

Comments and Suggestions for Authors

On the extraction rate and properties of humic acid, the effects of various process mechanical energies, KOH solution concentration, solid-liquid ratio, temperature, and time were examined. Specific suggestions are as follows: 

1. There is no source or reference for many calculation methods in this paper. For example, calculation of activation rate of mechanical energy activated weathered coal. 

2. The figure quotations did not meet the requirement. For example, figures 1 to 5 are not quoted in the corresponding text.

3. The paper holds that: “The free humic acid and total humic acid content decreased more under too high ball milling speed, because the impact of too much ball milling energy damaged the molecular structure of humic acid, which caused the destruction and decomposition of humic acid to a certain extent.” Humic acid is an alkaline extract, not a component of weathered coal, and the molecular structure of humic acid is assumed not destroyed by ball milling energy. You may need evidence to support your idea.  

4. The paper holds that: “……the weathered coal under the high rotational speed was darker in color, which indicated that the ball milling speed had a greater ability to affect the weathered coal than the ball milling time in the present test.” This conclusion seems groundless, because the high temperature caused by friction during grinding may change the color of the coal sample when the speed is increased. 

5. The authors may want to evaluate the reasonableness of all the experiment results. Did other researchers obtain similar results? There needs some comparison data from other studies to support your data. 

6. Similarly, in the discussion part, there is few evidence from other studies, thus it is hard to testify the soundness of your viewpoint.

7. The “conclusion” is missing in this manuscript.

Comments on the Quality of English Language

Minor editing of English language required.

Round 2

Reviewer 2 Report

Comments and Suggestions for Authors

There are some issues needs to be modified.

1. line 39, 44, Chinese full stop

2. What is the basis for the data in Table 1 and 2, air dry, dry, or dry ash free basis ?

3. Line 182-186, subscript for NaHCO3.

4. 397, Why there is “3.5. Humic acid heavy metal and toxic elements determination results” ?

Comments on the Quality of English Language

Minor editing of English language required.
